# Spatio-Temporal Variations and Driving Forces of Harmful Algal Blooms in Chaohu Lake: A Multi-Source Remote Sensing Approach

**Jieying Ma [1], Shuanggen Jin [1,2], Jian Li [1,\*], Yang He [2] and Wei Shang [2]**

1 School of Remote Sensing and Surveying Engineering, Nanjing University of Information Science and Technology, Nanjing 210044, China; mjy@nuist.edu.cn (J.M.); sgjin@shao.ac.cn (S.J.)
2 Shanghai Astronomical Observatory, Chinese Academy of Sciences, Shanghai 200030, China; heyang2014@whu.edu.cn (Y.H.); shangwei@shao.ac.cn (W.S.)
\* Correspondence: lijian@nuist.edu.cn; Tel.: +86-25-58235371

**Abstract:** Harmful algal blooms (hereafter HABs) pose significant threats to aquatic health and environmental safety. Although satellite remote sensing can monitor HABs at a large-scale, it is always a challenge to achieve both high spatial and high temporal resolution simultaneously with a single earth observation system (EOS) sensor, which is much needed for aquatic environment monitoring of inland lakes. This study proposes a multi-source remote sensing-based approach for HAB monitoring in Chaohu Lake, China, which integrates Terra/Aqua MODIS, Landsat 8 OLI, and Sentinel-2A/B MSI to attain high temporal and spatial resolution observations. According to the absorption characteristics and fluorescence peaks of HABs on remote sensing reflectance, the normalized difference vegetation index (NDVI) algorithm for MODIS, the floating algae index (FAI) and NDVI combined algorithm for Landsat 8, and the NDVI and chlorophyll reflection peak intensity index ($\rho_{chl}$) algorithm for Sentinel-2A/B MSI are used to extract HAB. The accuracies of the normalized difference vegetation index (NDVI), floating algae index (FAI), and chlorophyll reflection peak intensity index ($\rho_{chl}$) are 96.1%, 95.6%, and 93.8% with the RMSE values of 4.52, 2.43, 2.58 km$^2$, respectively. The combination of NDVI and $\rho_{chl}$ can effectively avoid misidentification of water and algae mixed pixels. Results revealed that the HAB in Chaohu Lake breaks out from May to November; peaks in June, July, and August; and more frequently occurs in the western region. Analysis of the HAB's potential driving forces, including environmental and meteorological factors of temperature, rainfall, sunshine hours, and wind, indicated that higher temperatures and light rain favored this HAB. Wind is the primary factor in boosting the HAB's growth, and the variation of a HAB's surface in two days can reach up to 24.61%. Multi-source remote sensing provides higher observation frequency and more detailed spatial information on a HAB, particularly the HAB's long-short term changes in their area.

**Keywords:** HAB; multi-source remote sensing; MODIS; Landsat; sentinel; Chaohu Lake

## 1. Introduction

As a vital freshwater resource, lakes provide essential and diverse habitats and ecosystem functions, and play vital roles in climate regulation, global carbon, nutrient cycles, thereby contributing to the industrial, agricultural, and food industries around the lakes [1]. However, the aquatic environment has been put at risk by both climate change and anthropogenic factors [2,3]. Wastewater discharge, farmland drainage, soil erosion, and agricultural fertilization are also primary nutrient sources leading to lake eutrophication. Besides, nitrogen and phosphorus pollution from inefficient sewage treatment systems and agricultural practices threaten to increase pollution and cause inland lakes' eutrophication [4]. Lake eutrophication may cause a harmful algal bloom (HAB), which is widely distributed, adaptable, and destructive [5]. A HAB increases oxygen consumption in the

water, releases toxins, degrades the water quality, and critically affects drinking water safety [4]. Comprehensive monitoring of HAB is vital in governing and repairing the lake environment [6], which has recently attracted more attention from both governments and the academic community.

Since both the environmental and meteorological factors may influence the breakout and spread of a HAB, it is important to study how these driving factors affect the HAB for effective management. Environmental factors, including the nutrients in water from fertilizer, agricultural nitrogen fixation, grain nitrogen, and feed nitrogen, are the primary sources of lake eutrophication [7–12], which have certain effects on HAB growth. Iron is an important component of the nitrate and nitrite reductase system, and its effect on enhancing the reduction efficiency and transfer rate of nitrate substances by algae is very observable [13]. Meteorological factors, including temperature, wind speed, precipitation, sunshine hours, etc., are also vital in HAB breakout. Previous research proved that the growth of cyanobacteria was directly proportional to the water temperature when greater than 18 °C, and that the activity of microcystis decreased when the temperature was greater than 30 °C [2], and HABs mostly occur in summer with proper temperature and sunshine hours. Variations in rainfall lead to a significant increase in nitrogen, which may lead to a HAB [14]. However, the influences of these factors on HAB are varied in different lakes, which requires further research in the region of interest.

It is challenging to capture the HAB dynamics using a conventional field sampling method due to the significant spatial-temporal variations of HAB [15]. Satellite remote sensing has been extensively used for monitoring the spatial coverage and temporal trends of HAB [16]. Many HAB detecting methods, including visual interpretation, supervised classification [17], single-band threshold [18], the spectral index method [19], and the water quality inversion method [16] have been developed. The visual interpretation delineates the HAB distribution using false-color composite satellite images [20], which is high-precision but low efficiency and is prone to personal misjudgment. The single threshold or spectral index methods, such as the normalized cyanobacteria index (NDI_CB) for Landsat-7 ETM+ [21] and *FAI* for Terra/Aqua MODIS, apply a single threshold for single or multiple bands data for HAB detection, which is simple and easy to implement [20,22]. Moreover, some research uses algal or chlorophyll concentration derived from remote sensing images to monitor HAB [23]. For example, the HABs were identified using chlorophyll inversion models on SeaWiFS from 1988 to 2002, on the Korean coast [24]. However, the uncertainties of these methods depend on regional water properties, sensor selection, and a threshold determination, which thus requires comprehensive assessments for method selection and implementation.

Among existing satellite images, Terra/Aqua MODIS imagery has been preferred for HAB monitoring due to its high temporal and spectral resolution [25]. However, the capabilities of Terra/Aqua MODIS are still limited by the low spatial resolution (250/500/1000 m), making it different to identify HABs in small and medium inland lakes [26]. For example, the optimal spatial resolution to monitor HAB in the Great Lakes is at most 50 m [27]. The Landsat TM/ETM+/OLI provides a higher spatial resolution (30 m), but its low revisit period (16 days) cannot track HAB's variations over time [22]. Sentinel-2A/B satellites launched on 23 June 2015 by the European Space Agency have wider spatial coverage and higher temporal resolution for monitoring of HABs [28]. Therefore, there is a pressing need for an effective and practical approach to capturing spatio-temporal variability of inland lake HAB integrating multi-source remote sensing techniques, which involves determining the appropriate algorithm and threshold for varied satellite sensors, and integration of HAB results.

Given this background, in this paper, multi-satellite images, including Sentinel-2A, Landsat 8 OLI, and Terra/Aqua MODIS, are used to monitor the spatial and temporal variations of HAB in Chaohu Lake, mostly its short-term variations. The proper algorithm was evaluated and adopted for different satellite sensors, and the accuracy and uncertainty

were analyzed. Based on HAB results from multi-source data, the variations and driving forces of HAB in Chaohu Lake for environmental management are discussed.

## 2. Study Area and Data

### 2.1. Study Area

Chaohu Lake, located in Hefei City, Anhui Province, is the fifth largest freshwater lake in China (Figure 1, projection: Gauss—Kruger projection, geographic coordinate system: World Geodetic System 1984). The tributaries of Chaohu Lake mainly include the Nanfei River, Shiwuli River, Pai River, Hangbu River, Baishitian River, Zhao River, Yuxi River, and Shuangqiao River. Chaohu Lake has an inflow of 344.2 million m$^3$ and an outflow of 23 million m$^3$. The center of Chaohu Lake is located at 29°47′–31°16′ north and 115°45′–117°44′ east, with an average water depth of 2.89 m and an average annual lake temperature of about 20 °C [29]. The terrain around the lake is mostly mountains and hills, and the Chaohu Lake basin is cultivated mainly by rice, wheat, rape, cotton, and corn. The agricultural land around the lake makes it easily accumulate nutritive salt in the water, causing severe non-point source pollution, which caused the lake's external pollution load, mainly originating from the northwestern part of the basin [30,31]. Nutrients in farmland are mainly composed of phosphorus and nitrogen, and the inflow of total phosphorus and total nitrogen is one of the main reasons for the eutrophication of Chaohu Lake. Chaohu Lake has become one of the most eutrophic lakes in China [32]. The total phosphorus concentration was one of the main driving factors affecting Anabaena and microcystins' spatial and temporal distribution [33,34]. The farming period is from June to November. The average annual rainfall in Chaohu Lake is 224 mm, which drives the farmland nutrients to the lake during the farming period [35]. Moreover, the rain stirs up the mud at the bottom of Chaohu Lake, and large amounts of nutrient salts in the mud turn up, increasing the concentration of nutrient salts in Chaohu Lake. The total phosphorus content in Chaohu Lake is 0.131mg/L, and the total nitrogen content is 2.04 mg/L. The nitrogen and phosphorus ratio of optimum reproduction of the dominant species of HAB in Chaohu Lake was about 11.8:1 [36]. According to the monitoring data over the years, the ratio of nitrogen to phosphorus in Chaohu Lake is between 10:1 and 15:1, resulting in an outbreak situation of non-point source HAB [37]. When algae proliferate and die, they accelerate the consumption of dissolved oxygen in water, resulting in the death of many aquatic animals and plants, weakening the purification capacity of water, and causing severe harm to human health [5]. Therefore, it is essential to monitor the water environment with joint multi-source remote sensors.

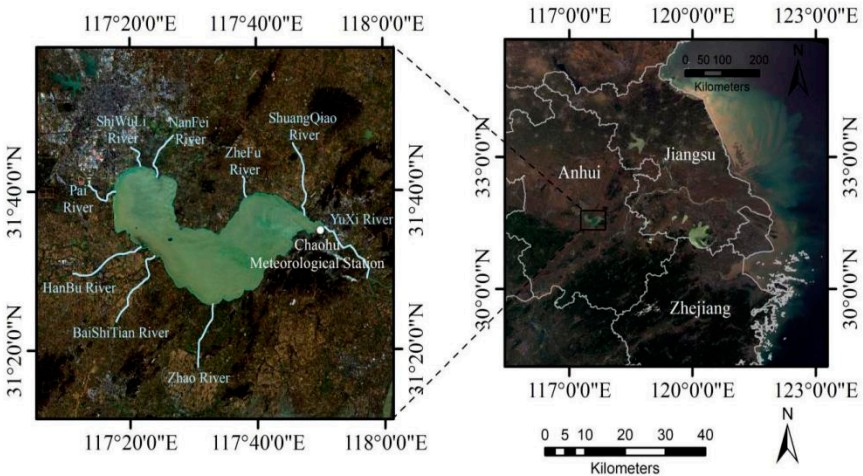

**Figure 1.** Location of Chaohu Lake.

### 2.2. Remote Sensing Data

A total of 420 images of Terra/Aqua MODIS L-1B data (MOD02) in 2019 were selected and downloaded from Earthdata's website (https://search.earthdata.nasa.gov/). Two images of Landsat 8 OLI (Level 1) were downloaded from the USGS official website of shared data (https://earthexplorer.usgs.gov/). A total of 16 images of Sentinel-2 MSI satellite data (L1C) were downloaded from the official website of ESA (https://scihub.copernicus.eu/). Clear and cloudless images were picked out (see Table 1) and preprocessed, including re-projection and geometric correction. Figure 2 shows the different cloudless products distributed in the space in 2019 so one can picture the time lag between the different satellite acquisitions.

**Table 1.** Multi-sensor data of the cloudless images of Chaohu Lake in 2019.

| 2019 | Resolution | Revisit Period | May | June | July | August | September | October | November |
|---|---|---|---|---|---|---|---|---|---|
| Terra/MODIS | 250 m | 1 day | 4 | 13 | 10 | 16 | 12 | 12 | 12 |
| Aqua/MODIS | 250 m | 1 day | 2 | 3 | 3 | 2 | 3 | 5 | 5 |
| Landsat8 OLI | 30 m | 16 days | 0 | 0 | 0 | 2 | 0 | 0 | 0 |
| Sentinel-2A MSI | 20 m | 10 days | 2 | 1 | 2 | 0 | 2 | 3 | 1 |
| Sentinel-2B MSI | 20 m | 10 days | 0 | 1 | 0 | 0 | 0 | 2 | 2 |
| Total | - | - | 8 | 17 | 13 | 20 | 17 | 19 | 18 |

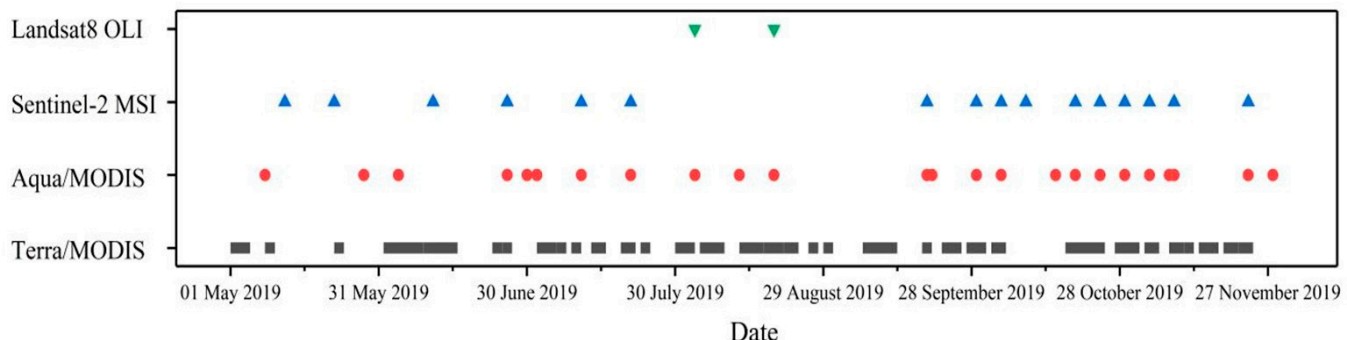

**Figure 2.** Annual distribution of cloudless images from multi-sensor data.

### 2.3. Environmental and Meteorological Data

The meteorological analysis data were obtained from the Meteorological Center of the National Meteorological Administration (http://www.cma.gov.cn/) (Figure 3). In 2019, Chaohu Meteorological Station's maximum sunshine hours, maximum temperature, and maximum wind speed occurred in May, July, and August, respectively. The variation range of wind speed was 0.5–6.4 m/s, the maximum number of sunshine hours was 12.9 h, and the time of direct sunlight was half a day. The average rainfall was 224 mm. The average maximum temperature was 33.9 °C.

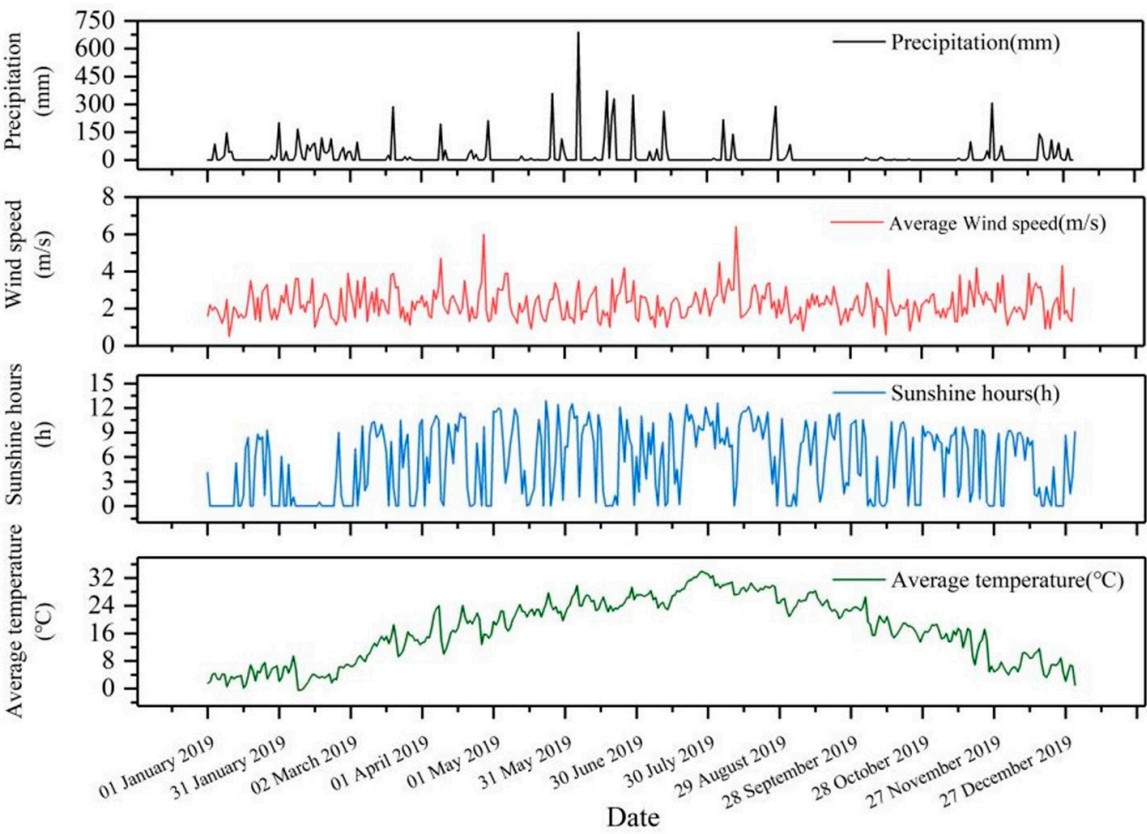

**Figure 3.** The variation diagram of rainfall, sunshine hours, average temperature, and precipitation in Chaohu Station in 2019.

## 3. Methods

Figure 4 is the technical flow chart of this paper, using which the original satellite data were obtained and preprocessed. The most appropriate algorithms were selected respectively for Sentinel-2 MSI, Terra/Aqua MODIS, and Landsat 8 OLI to obtain the distribution map of HAB in Chaohu Lake, and we checked the accuracy of the algorithms with visual interpretation results. Finally, the formation and distribution of HAB were analyzed by combining various meteorological factors.

### 3.1. Data Preprocessing

The preprocessing steps mainly included geometric correction, radiometric calibration, and atmospheric correction. Landsat-8 OLI and Terra/Aqua MODIS data were preprocessed using ENVI software (ENVI 5.3) to convert DN (digital number) values into TOA (top of atmosphere reflectance) radiance or reflectance after radiometric calibration, and then different atmospheric correction models were selected according to different data sources. The FLAASH atmospheric correction module (Fast Line-of-sight Atmospheric Analysis of Spectral Hypercubes) was adopted for Landsat 8 OLI, which was based on the MODTRAN-4 (Moderate Spectral Resolution Atmospheric Transmittance Algorithm and Computer Model) radiation transmission model, with high accuracy. It can maximally eliminate the influences of water vapor and aerosol scattering over case II waters, and has been successfully used in previous studies from Landsat 8 OLI [38,39]. MODIS images were atmospherically corrected using the dark-objects method [40–42]. The procedure was to select the relatively clean area as a region of interest in the eastern part of Chaohu Lake, and statistically analyze the pixel brightness value of each band, while using a non-zero pixel with a suddenly increased brightness value as the dark pixel value. The selected dark pixel was used as the distance luminance value for atmospheric correction. Sentinel-2A/B original L-1C images were mainly processed using SEN2COR (version: Sen2Cor-02.08.00-win64)



for radiometric calibration and atmospheric correction. SEN2COR is a plug-in released by the European Space Agency (ESA) specifically for Sentinel-2 atmospheric calibration. The spectral curve of the image by SEN2COR with atmospheric correction of Sentinel-2 images is consistent with the trend of the actual spectral curve on the ground [43]. The reflectance after atmospheric correction was compared with the field spectra of 39 ground objects; $R^2$ was 0.82 and the root mean square error was 0.04 [44], indicating high accuracy. All the images selected in the experiment were mostly cloudless. Before determining the HAB, cloud-covered regions of the remote sensing images were made into a cloud mask product by the single-band threshold method to eliminate the influence of clouds [45].

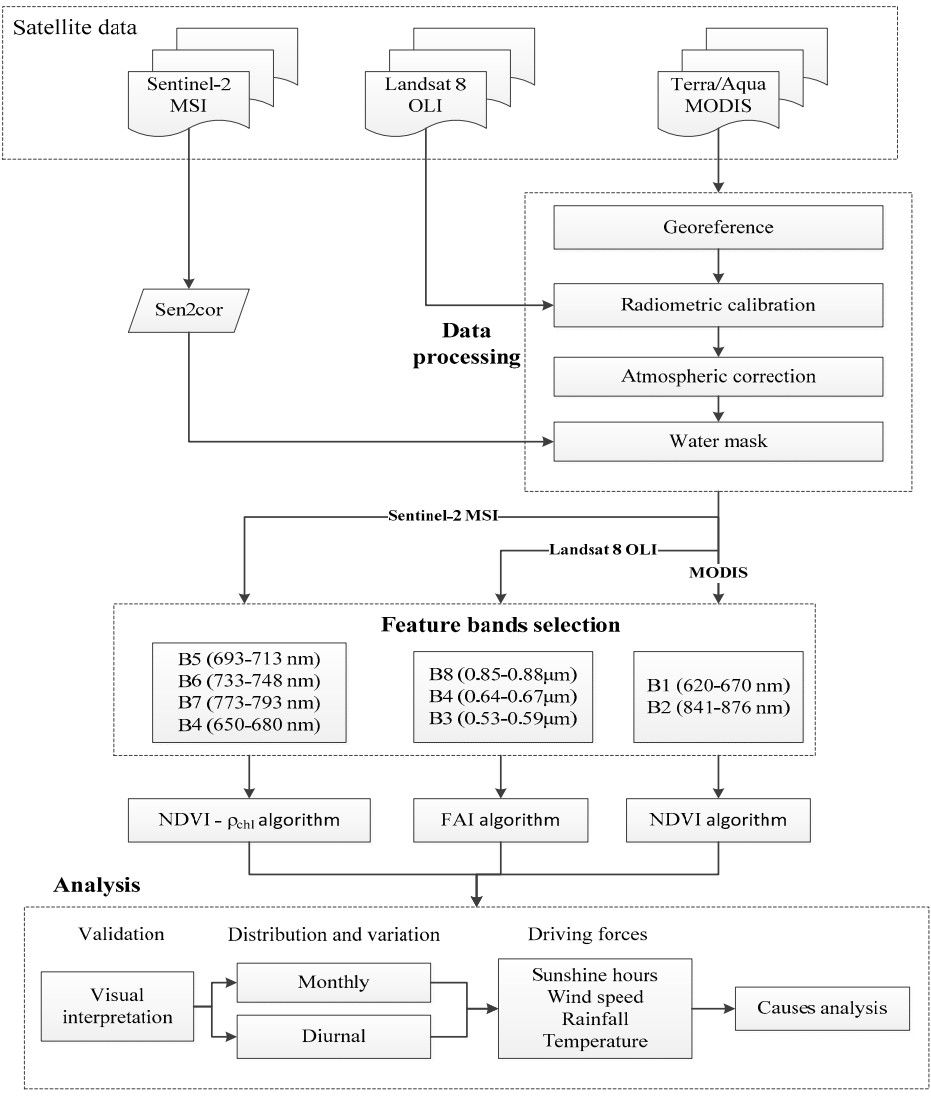

**Figure 4.** Technical route flow chart.

### 3.2. Extraction Algorithm of HAB

Algae in water would cause an absorption peak near the wavelength of 620–630 nm and a reflection peak at 650 nm, with a sharp increase in reflectance at around 700 nm [46]. High absorption in the red band by vegetation pigments and high reflection in the near-infrared band have been used for a long time to detect vegetation coverage, and eliminate some radiation errors. NDVI can reflect the background influence of the vegetation canopy. Therefore, the NDVI algorithm of MODIS was used for monitoring HAB in Chaohu Lake [47]. RGB band synthesis of Landsat/OLI B8 (0.85–0.88 μm), B4 (0.64–0.67 μm), and B3 (0.53–0.59 μm) renders HABs in a reddish color, in strong contrast with the bloom-free dark

water, making it easy do distinguish bloom and non-bloom areas. Due to the influences of lake currents and wind, HAB areas generally present as elongated strips [48,49]. The *FAI* algorithm can eliminate the impact of the atmosphere by using the combination of these three bands. Compared with NDVI algorithm easily influenced by the observation environment, *FAI* would be suitable for the Landsat images. Unlike MODIS and Landsat 8, Sentinel-2 MSI was equipped with multiple spectral bands and 20 m ground resolution. Three special bands, B5 (693–713 nm), B6 (733–748 nm), and B7 (773–793 nm), are set for vegetation monitoring, which is also sensitive for HABs [50,51]. Therefore, the $\rho_{chl}$-NDVI algorithm is used for improving the accuracy of acquiring HAB in Chaohu Lake by fusing these 5 characteristic bands. Detailed descriptions of these algorithms are included in Figure 5.

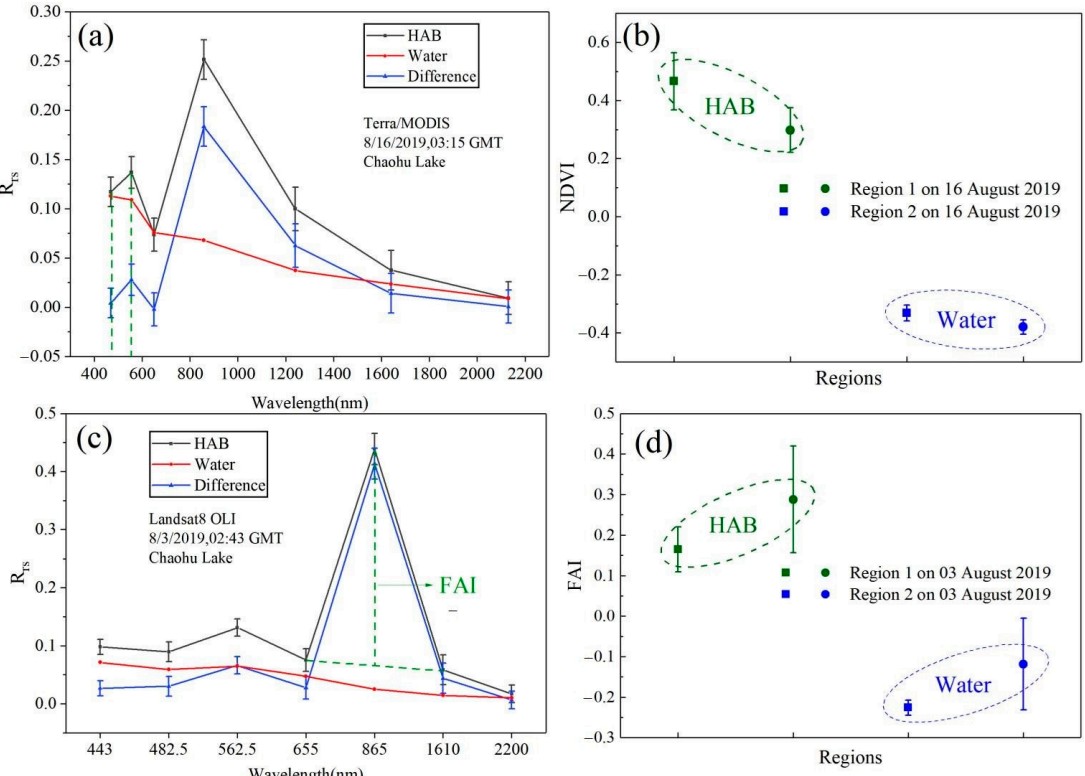

**Figure 5.** The interpretation of the algorithm and the results of each algorithm's reflectivity diagram (**a**,**c**,**e**) are the reflectances of a harmful algal bloom (HAB) and a nearby non-HAB lake, and (**b**,**d**,**f**) are means and standard deviations of HAB and non-HAB water lake reflectance. For the two regions of MODIS, Landsat8, and Sentinel-2 150 × 150 image pixels with 9359 × 9459, 7651 × 7791, and 0980 × 10,980 HAB classification pixels respectively.

### 3.2.1. Normalized Vegetation Index (NDVI)

Rouse [52] first used Landsat-1 MSS data to propose a NDVI based on the characteristic that the reflectivity of all vegetation increases dramatically near 700 nm. NDVI can reflect surface vegetation coverage [53]. Therefore, as the most common method, NDVI has been widely used in the study of algal extraction [54–56], which can eliminate the influences of terrain, shadow, and solar elevation angle [57]:

$$NDVI = \frac{\rho NIR - \rho RED}{\rho NIR + \rho RED} \tag{1}$$

where $\rho_{RED}$ and $\rho_{NIR}$ represent the reflectances of the red band and near-infrared band.

### 3.2.2. Floating Algae Index (FAI)

The floating algae index was first proposed by Hu [58]. *FAI* is defined as a linear spread of reflectivity in the near-infrared, red, and short-wave infrared regions, and can be

applied to monitor proliferating algae, such as Ulva or Sargassum spp [59]. The observation results of this algorithm provide strong robustness. *FAI* is less affected by atmospheric environment, observation conditions, and water reflectivity absorption in the near-infrared band [60]. *FAI* is often used to identify dense HABs in marine and inland waters [61]. Therefore, the spectral information of the red band, near-infrared band, and short-wave infrared band can be used to correct the atmospheric effects [35]. The algorithm is as follows:

$$FAI = R_{NIR} - R'_{NIR} \tag{2}$$

$$R'_{NIR} = R_{RED} + (R_{SWIR} - R_{RED}) \times \frac{\lambda_{NIR} - \lambda_{RED}}{\lambda_{SWIR} - \lambda_{RED}} \tag{3}$$

where $R_{RED}$, $R_{NIR}$, and $R_{SWIR}$ represent the reflectances of red, near-infrared, and short-wave infrared bands respectively; $\lambda_{RED}$, $\lambda_{NIR}$, and $\lambda_{SWIR}$ represent the central wavelengths; and $R'_{NIR}$ is the interpolating reflectance—namely, the reflectivity information of the infrared band can be obtained by linear interpolation of the red band and the short-wave infrared band.

The gradient contrast method was used for *FAI* algorithm to determine the threshold of HAB. The experimental results showed that $FAI < -0.01$ and $FAI > 0.02$ were non-bloom regions [19]. According to the average threshold value of the gradient diagram, $FAI > -0.002$ was finally determined to be the region of HAB.

### 3.2.3. Chlorophyll Reflection Peak Intensity Algorithm

Algae also contain chlorophyll, like land plants, so when the algae aggregates, the spectrum shows a vegetation-like characteristic [62,63]. Chlorophyll shows troughs at 420–500 nm (blue and violet light band) and 625 nm, and a small peak value is found at the central wavelength of the green band [36]. Based on the correlation between algae and chlorophyll concentration, the following model was constructed to identify the concentration of HAB [37,64]:

$$\rho_{chl} = \rho(560) - \frac{\rho(490) + \rho(665)}{2} \tag{4}$$

where $\rho(490)$, $\rho(560)$, and $\rho(665)$ correspond to the reflectivity of the blue, green, and vegetation red edge bands of the Sentinel-2A satellite.

### 3.3. Accuracy Assessment

To obtain the reference data or "truth data" for accuracy assessment of HAB detection from different satellite data, the visual interpretation method was used on false-color images. The verification data of the spatial distribution and area statistics of HAB were also obtained from the Department of the Ecological Environment of Anhui Province (http://sthjt.ah.gov.cn/), which have been checked through ground monitoring points, field investigations, and validation. The root mean square error (RMSE) and relative error (RE) were used to evaluate the accuracy of the HAB extractions using the NDVI algorithm. Additionally, the accuracies of different HAB detection methods were assessed using following indexes [17]:

Correct extraction rate (*R*) is the percentage of the extracted HAB area over the true data:

$$R = \frac{A_r}{A_{truth}} \times 100\% \tag{5}$$

Over-extraction rate (*W*) is the percentage of mixed extracted HAB area over the true data:

$$W = \frac{A_w}{A_{truth}} \times 100\% \tag{6}$$

Omitted extraction rate ($M$) is the percentage of the unextracted HAB area over the truth data:

$$M = \frac{A_{\text{m}}}{A_{\text{truth}}} \times 100\% \tag{7}$$

The reference data of HAB were denoted as $A_{\text{truth}}$. The area statistic of HAB extracted by various extraction methods was designated as $A$. The overlapping part of $A$ and $A_{\text{truth}}$ was regarded as the correct extracted part, which was denoted as $A_{\text{r}}$. The disjoint part of $A$ is considered to be the extracted by mistake, which was denoted as $A_{\text{w}}$. In $A_{\text{truth}}$, the disjoint part was regarded as the missing part, which was denoted as $A_{\text{m}}$.

## 4. Results

Visual interpretation was analyzed based on 86 MODIS images and 2 Landsat images; 16 Sentinel-2 images were used to be the verification data to compare the accuracy of each extraction algorithm (Figure 6).

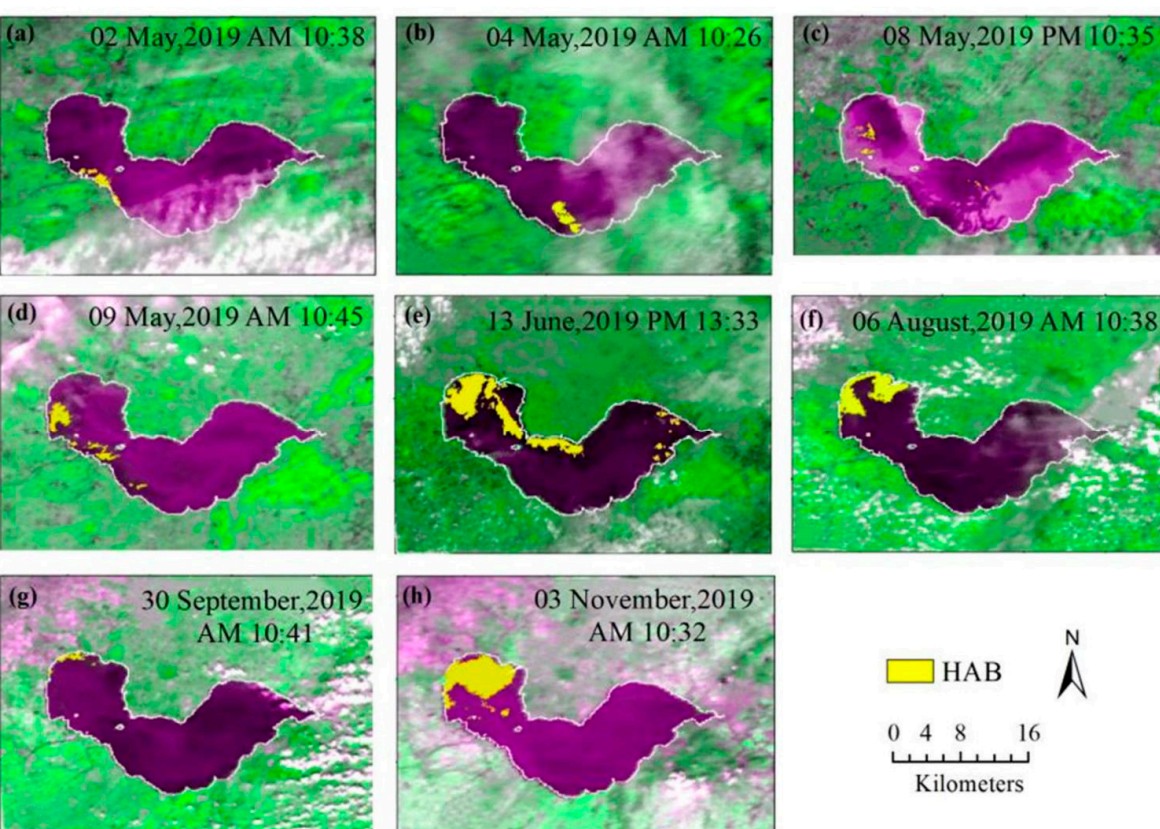

**Figure 6.** Spatial distribution of HAB in Chaohu Lake from visual interpretation.: (**a**) HAB distribution at 10:38 on 2 May, (**b**) HAB distribution at 10:26 on 4 May, (**c**) HAB distribution at 10:35 on 8 May, (**d**) HAB distribution at 10:45 on 9 May, (**e**) HAB distribution at 13:33 on 13 June, (**f**) HAB distribution at 10:38 on 6 August, (**g**) HAB distribution at 10:41 on 30 September, (**h**) HAB distribution at 10:32 on 3 November.

### 4.1. Accuracy of HAB Algorithms

Depending on the algorithm selection and analysis in Section 3.2, NDVI was used for MODIS to extract HAB. The comparison of NDVI and $\rho_{\text{chl}}$ values showed that for a low concentration of HAB, the threshold for $\rho_{\text{chl}}$ was 0.05, and the NDVI threshold was 0.24. For a moderate or high algae concentration, the threshold for $\rho_{\text{chl}}$ was 0.09, and NDVI was larger than 0.68. Therefore, a pixel with an NDVI > 0 was first classified as a vegetation pixel, and then combined with $\rho_{\text{chl}} > 0.05$ was judged as belonging to a HAB. NDVI < 0 and $\rho_{\text{chl}} > 0.03$ was an "algal-water" suspension region and also judged as HAB.

The RMSE was 4.27 km$^2$ and RE was 15.9% when compared to HAB products reached by visual interpretation (Figure 7). For the significance test, $p < 0.05$, the results showed that the HAB region observed by satellite was consistent with the visual interpretation. Residual normal distribution of HAB areas extracted by MODIS and Sentinel-2 was showed on Figure 8, R$^2$ was 0.98 and 0.99 between MODIS, Sentinel-2 and visual interpretation, respectively. The Sentinel-2 MSI, MODIS, and Landsat 8 OLI randomly selected the day of the HAB outbreak, and a confusion matrix was used to evaluate the classification accuracy between the monitoring results and the visual interpretation (Table 2).

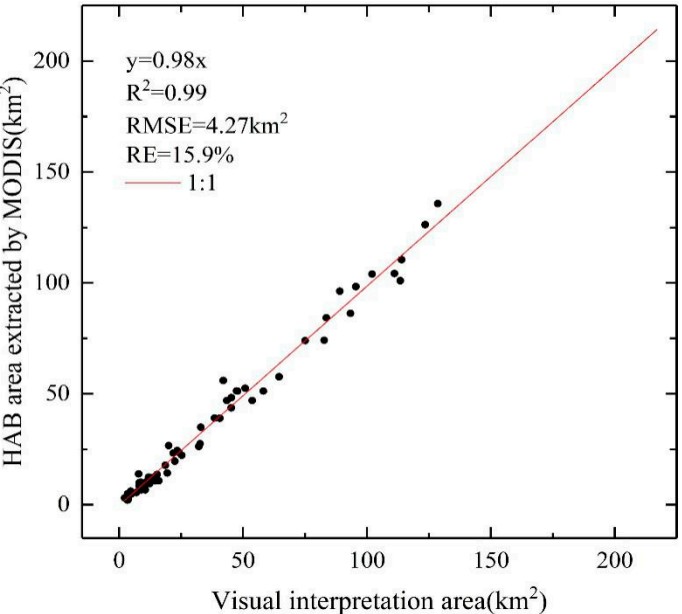

**Figure 7.** Comparison of HAB extracted by MODIS and visual interpretation.

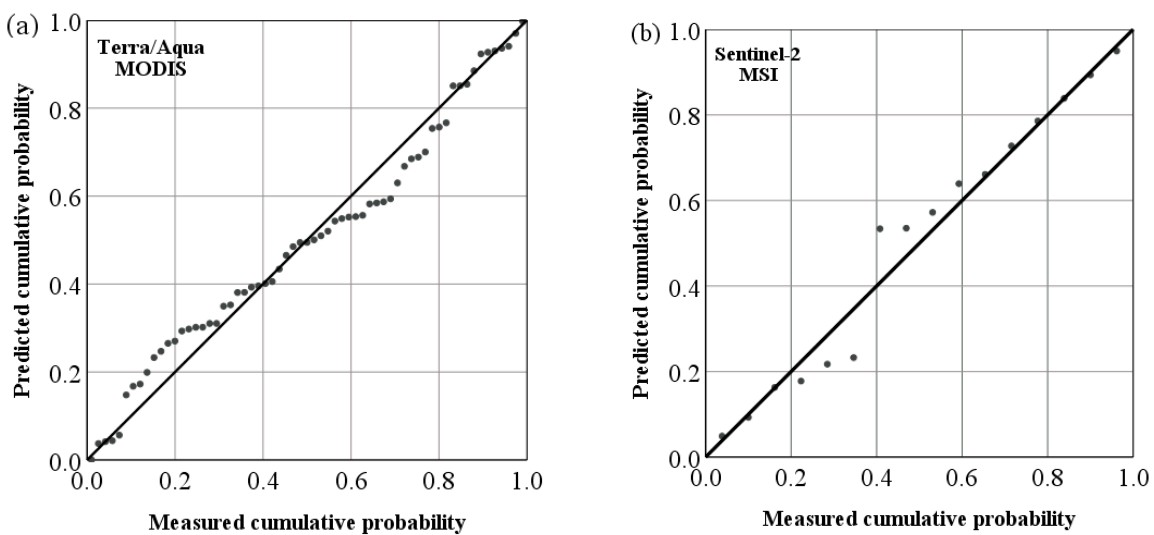

**Figure 8.** Residual normal distribution of HAB areas extracted by MODIS and Sentinel-2: (**a**) the coefficient of determination (R$^2$) was 0.98 between MODIS and visual interpretation, and (**b**) R$^2$ was 0.99 between Sentinel-2 and visual interpretation.

**Table 2.** The confusion matrix between the extraction results and visual interpretation.

| | Class | HAB | Water | Cloud | Total | Accuracy |
|---|---|---|---|---|---|---|
| Sentinel-2 MSI 22 May 2019 | HAB | 78.47 | 0.02 | 0.00 | 0.92 | Overall Accuracy = (17,105,160/17,189,550) 99.5091% Kappa Coefficient = 0.9002 |
| | Water | 20.57 | 99.76 | 0.26 | 97.49 | |
| | Cloud | 0.96 | 0.23 | 99.73 | 1.58 | |
| | Total | 100.00 | 100.00 | 100.00 | 100.00 | |
| Landsat 8 OLI 19 August 2019 | HAB | 95.93 | 0.01 | 0.53 | 1.62 | Overall Accuracy = (1,907,160/1,909,950) = 99.8539% Kappa Coefficient = 0.9972 |
| | Water | 4.07 | 99.99 | 8.07 | 97.51 | |
| | Cloud | 0.00 | 0.00 | 91.40 | 0.77 | |
| | Total | 100.00 | 100.00 | 100.00 | 100.00 | |
| Terra/MODIS 1 August 2019 | HAB | 93.86 | 0.00 | 18.29 | 2.71 | Overall Accuracy = (6124/6237) 98.1882% Kappa Coefficient = 0.8605 |
| | Water | 6.14 | 100.00 | 12.98 | 93.55 | |
| | Cloud | 0.00 | 0.00 | 68.73 | 3.74 | |
| | Total | 100.00 | 100.00 | 100.00 | 100.00 | |

NDVI and *FAI* were combined to detect HAB using Landsat 8 OLI images; NDVI and $\rho_{chl}$ were combined for Sentinel-2 MSI data. Table 3 shows the accuracy evaluation results when compared to visual interpretation products, which demonstrated that HAB extracted by NDVI and *FAI* has a relatively correct extraction rate of about 95%. The RMSE of HAB from *FAI* algorithm was 0.56 km$^2$ and RE was 3.9%. However, the NDVI extraction method was affected by thin cloud or fog, and the cloud shadow was misidentified as a HAB. Moreover, NDVI method may miss pixels with lower algae concentrations, when compared with FAI. By comparing the extraction results on 3 August 2019 and 19 August 2019, the over-extraction areas of the NDVI method due to the mixed pixels and clouds were found to be 1.46 and 0.18 km$^2$, respectively. A comprehensive comparison shows that the extraction of HAB by the two methods was consistent, but the *FAI* method was better than NDVI at the details. Better results were obtained by combining NDVI with the chlorophyll reflection peak $\rho_{chl}$, especially for regions with lower concentrations of HAB. According to this method, the correct extraction rate of the Sentinel-2 data reached 96.01%, while RMSE and RE were 2.4 km$^2$ and 6.2%, respectively.

**Table 3.** Accuracy for HAB extraction of Landsat 8 OLI, Sentnel-2 MSI, and Terra/Aqua MODIS data.

| | Extraction Method | Extracted Area (km$^2$) | Omission Area (km$^2$) | Overestimated Area (km$^2$) | Correct Area (km$^2$) | Missing Rate (%) | Over-Extraction Rate (%) | Correct Rate (%) |
|---|---|---|---|---|---|---|---|---|
| 3 August 2019 | *FAI* | 16.30 | 0.02 | 3.31 | 12.98 | 0.12% | 25.49% | 99.88% |
| | NDVI | 16.98 | 0.52 | 4.49 | 12.48 | 3.97% | 34.57% | 96.03% |
| | Visual interpretation | 13.00 | | | | | | |
| 4 October 2019 | Sentinel | | 0.55 | 3.75 | 13.27 | 3.99% | 27.12% | 96.01% |
| | Visual interpretation | 13.82 | | | | | | |
| 3 November 2019 | MODIS | | 1.84 | 18.88 | 10.21 | 3.92% | 40.16% | 96.08% |
| | Visual interpretation | 47.02 | | | | | | |

## 4.2. Monthly Variations of HAB

MODIS images were mainly used to track monthly HAB changes in Chaohu Lake in 2019 with the advantage of its high temporal resolution. The HAB in Chaohu Lake occurs between May and November (Figure 9). The northwestern part of Chaohu Lake is more seriously polluted by algae than the eastern, and the area of HAB reaches its maximum in July. The monthly frequency map is the ratio of the number of outbreaks in each region and month to the total numbers of the whole lake. The distribution frequency map indicates the probability of a HAB outbreak in each region of Chaohu Lake. Although HAB breaks out sometimes in a small region, they often occur in the west of the lake. According to the frequency distribution of inter-month HAB, it is increased in June and remains high from



June to November. The highest outbreak frequency occurs in the northwestern part of the lake in October, and the peak of distribution frequency of Chaohu Lake in the eastern lake appears in June.

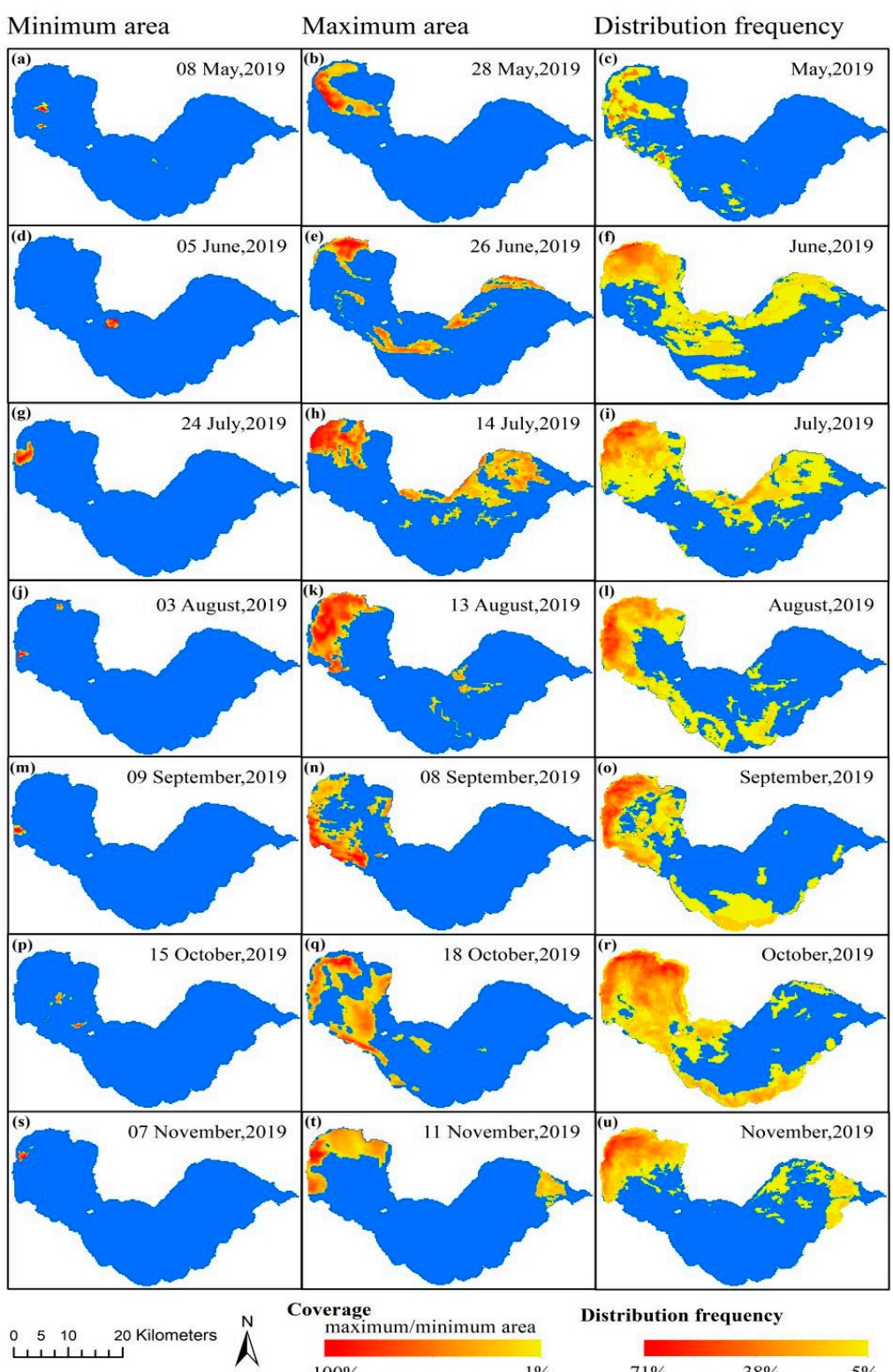

**Figure 9.** Spatial distribution of the minimum area, the maximum area, and the frequency of the monthly HAB in Chaohu Lake: (**a**,**d**,**g**,**j**,**m**,**p**,**s**) The minimum HAB area of each month from May to November, (**b**,**e**,**h**,**k**,**n**,**q**,**t**) The maximum HAB area of each month from May to November, (**c**,**f**,**i**,**l**,**o**,**r**,**u**) The HAB distribution frequency of each month from May to November.

The monthly coverage rates for the maximum, minimum, and average HAB area are shown in Figure 10. Adding up the maximum and the minimum area accounts for up to 50% of the total monthly HAB area in May, but the maximum HAB area was only 53.69 km². The average monthly coverage area was less than 20 km², which was the lowest in 2019. This indicates that the level of HAB in May was not serious. In contrast, from June to November, the maximum HAB area accounted for less than 25% to the total HAB area, and a HAB area exceeding 100 km² was always found in the mid-month. In July, the maximum area of HAB reached 217 km², accounting for 28.6% of the Chaohu Lake area, covering the northwestern and central parts of the lake. In 2019, the minimum HAB area was 1.625 km², which occurred on November 7, accounting for 0.2% of the total lake area. The average monthly coverage was lower than that in the period of HAB in Chaohu Lake (June to October). It indicated that the activity of HAB in Chaohu Lake began to decrease in November.

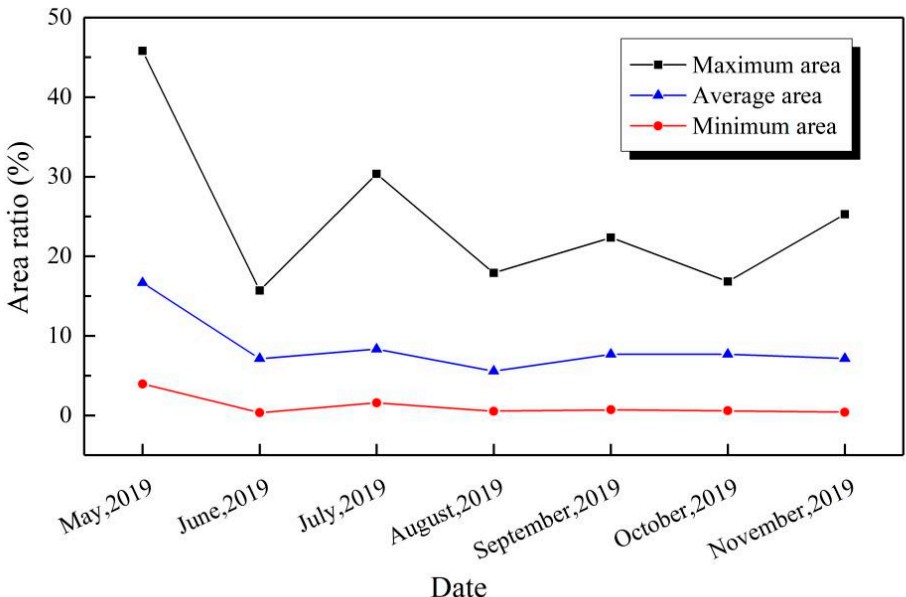

**Figure 10.** The ratios of the monthly maximum, minimum, and average HAB area to the total HAB area per month (total HAB area: monthly statistics of the area where HAB occurs each time).

### 4.3. Diurnal Variation of HAB

The spatial-temporal patterns of HABs can be easily affected by hydrology and meteorological factors, and thus induce dramatic variation in a short time, which requires high-frequency monitoring by the integration of a multi-satellite sensor. To reveal the diurnal variations of HAB in Chaohu Lake, multi-source satellite, including Sentinel-2 MSI, Landsat 8 OLI, and Terra/Aqua MODIS are integrated, as shown in Figure 11. While HAB is concentrated and stable, such as on 4 October 2019, the difference of extraction regions between Sentinel-2 MSI and Terra/MODIS is the smallest. Significant differences were observed due to the scattered distribution of HAB on June 26. In the surrounding areas with low algal density, MODIS had a lower spatial resolution; the result may be biased due to the mixed pixels. Since Terra/MODIS is the morning satellite, it passes through the equator from north to south at about 10:30 local time, and Aqua/MODIS is the afternoon satellite and passes through the equator from south to north at about 13:30 local time. According to the common influence of all factors, the monitored HAB area and distribution were different at different times of passing the territory. Besides, there will also be weather effects, such as the possibility of cloud cover in the afternoon compared with the morning in the study area, which will also have impacts on the extraction and identification of HAB.

The HAB diurnal changes from Landsat 8 and MODIS images on 19 August 2019 have no significant differences in the area and distribution. The morphology of HAB monitored

by Terra (Figure 12a) was different from that of Landsat8 (Figure 12b), which may be due to the low quality (cloud coverage) of Terra/MODIS images on 3 August 2019. HAB region was disturbed by thin clouds, which could not represent the real distribution pattern at that time. The reliability of this result was also verified by the distribution diagram of bloom morphology in an Aqua image (Figure 12a). Compared with the result of Landsat 8 image (Figure 12d), the Aqua image (Figure 12e) result on August 3 showed a decrease in the distribution of HAB and a concentration increase in the coverage center. As the Terra image on August 3 was covered by clouds and fog, Figure 12 does not show the HAB distribution in the morning.

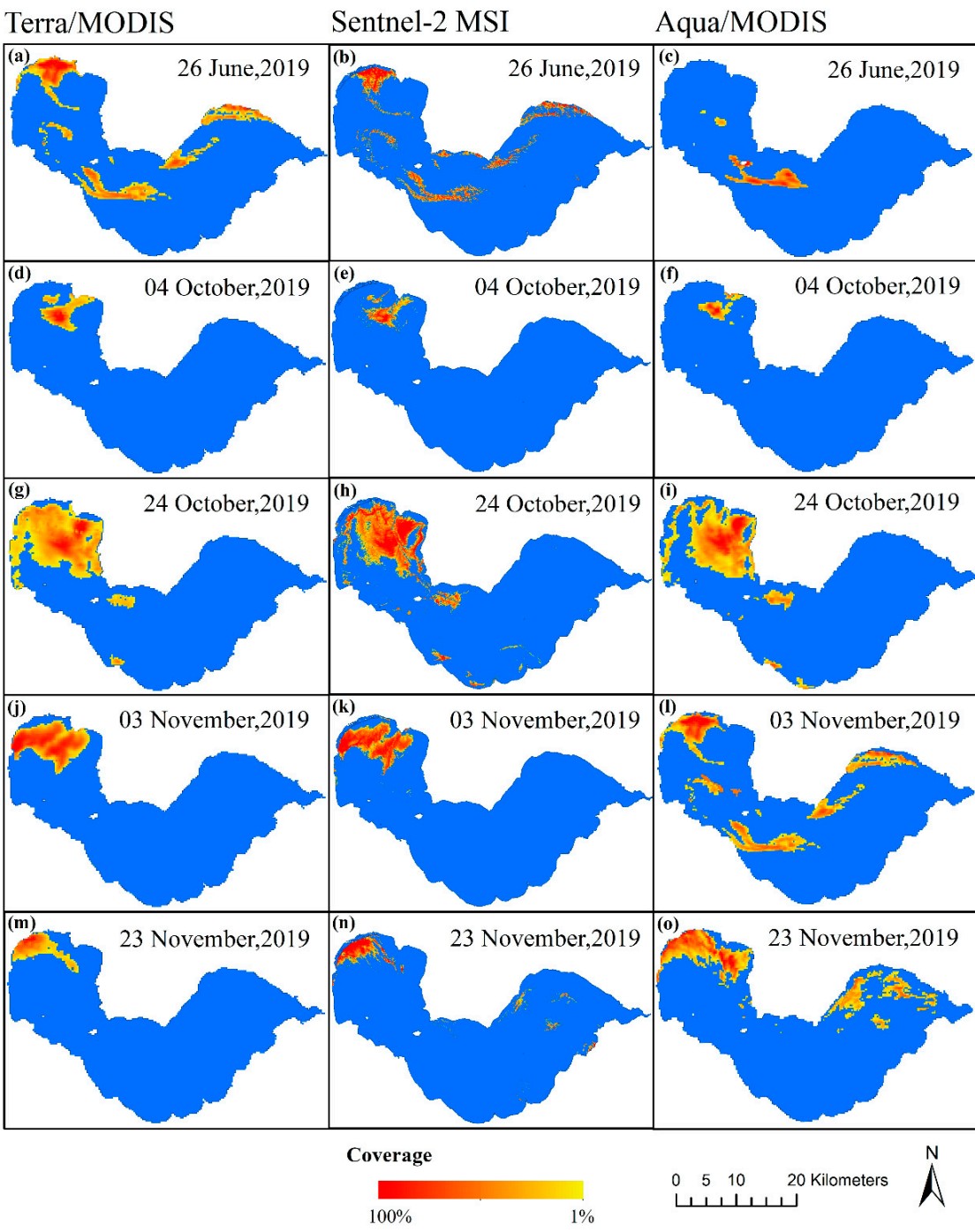

**Figure 11.** Variations of HAB from multi-satellite data: (**a,d,g,j,m**) results from Terra/MODIS, (**b,e,h,k,n**) results from Sentinel-2 MSI, (**c,f,i,l,o**) results from Aqua/MODIS.

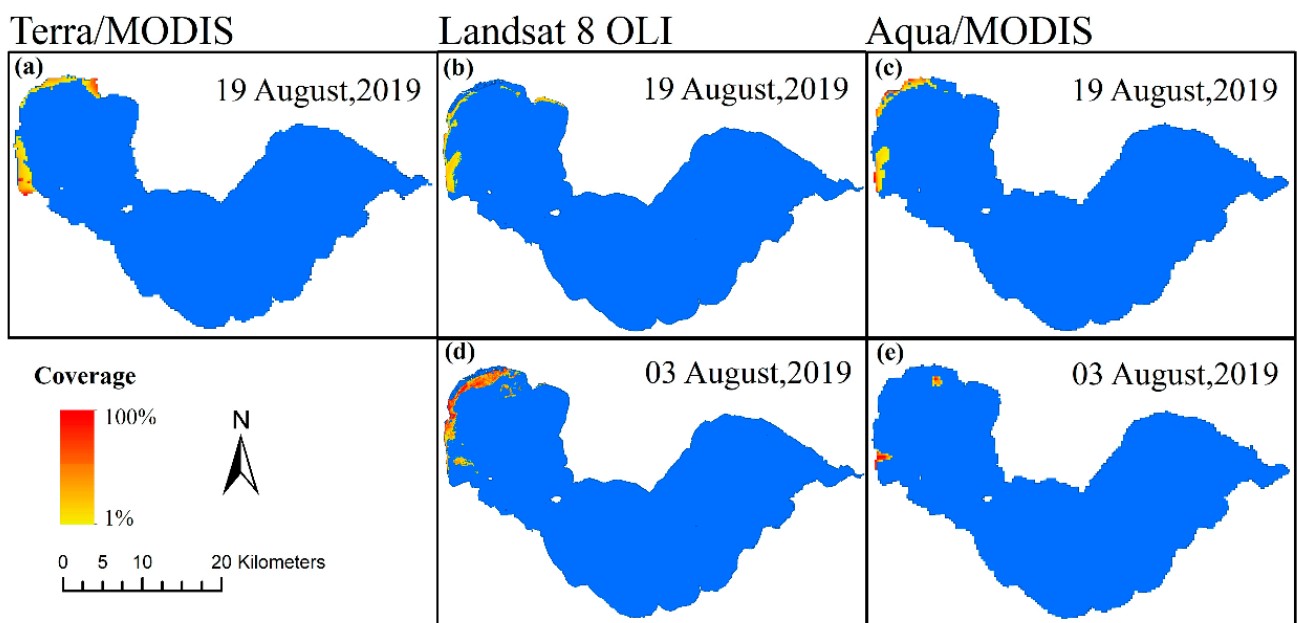

**Figure 12.** Harmful algal blooms from Terra/MODIS (**a**), Landsat 8 OLI (**b**,**d**), and Aqua/MODIS (**c**,**e**).

## 5. Discussion

### 5.1. Driving Forces of HAB

The driving forces for the breakout of HAB are of great concern for HAB control and management. Among many factors, the temperature, rainfall, sunshine hours, wind, radiation, etc., have drawn great attention [1]. Some previous research demonstrated that the degree of HAB is positively correlated with temperature, sunshine hours, and global radiation changes, and negatively correlated with precipitation and wind speed [65]. Our results showed similar results on the correlation between the HAB areas and both temperature and sunshine hours, but the $R^2$ was quite low (<0.05). However, increased temperature promotes the growth of HABs, and colder months may delay the occurrence of HAB [66]. It can be seen that the maximum and minimum areas of Chaohu Lake HAB in July were higher than in other months (Figure 13). The maximum, average, and minimum values of the HAB area in August and September were close. However, the number of hours of sunshine in September was 77.5 h lower than that in August. The low sunshine hours made it difficult for algae to reproduce and grow through photosynthesis, which inhibited the accumulation and explosion of large areas of HAB. However, too much sunshine will make algae inactive and also inhibit HAB growth. This is consistent with the conclusions from Zhang's research demonstrating that under high temperatures and with many sunshine hours, there will be no large-scale HAB [67,68]. Therefore, appropriate sunshine hours and temperature can promote the photosynthesis of algae.

The effect of precipitation showed a weak negative correlation with the HAB. The HAB on rainy days of August 3 and 7 was decreased by 79.3% and 61.3%, respectively, when compared with the previous days. This may indicate that the rainfall may dilute or inhibit the occurrence of HABs. HAB was often found on days after scattered rain, such as on 27 May, 26 August, and 17 October. In contrast, the total precipitation in September was half of that of August, and the scattered rain provided favorable conditions for the growth and reproduction of algae. Therefore, the rainfall was the main driving force of the monthly variations of the HAB from July to September. However, May–June rainfall is the highest and most frequent, which reduces the temperature of the water surface, and also reduces the density of algae and the concentrations of nutrients, making the probability of the occurrence of HABs only slightly increased in June compared with May. Rainfall decreased in July, the temperature increased, and the occurrence of HAB increased sharply. Therefore,

the low occurrence of HABs in June was caused by precipitation. Based on the analysis of previous data, it was found that the period of highest temperature is inconsistent with the month with the highest probability of HAB, and atmospheric temperature is the main meteorological factor affecting HAB [69,70]. From mid-July to mid-August, Chaohu Lake's temperature in 2019 reached its annual maximum and the average daily sunshine hours were all over 8 h. However, due to the hysteresis effect [71] of temperature on the response of HAB in Chaohu Lake, the precipitation mainly occurred from June to mid-July. Much rain in June transports the nutrients from the catchment area as the non-point source. The algae in July with the highest maximum area is due to the inflow during June. The effect of nutrient supply appears with a time lag because the controlling factor is temperature. Even with a high concentration of nutrients, insufficient temperature regulates blooming.

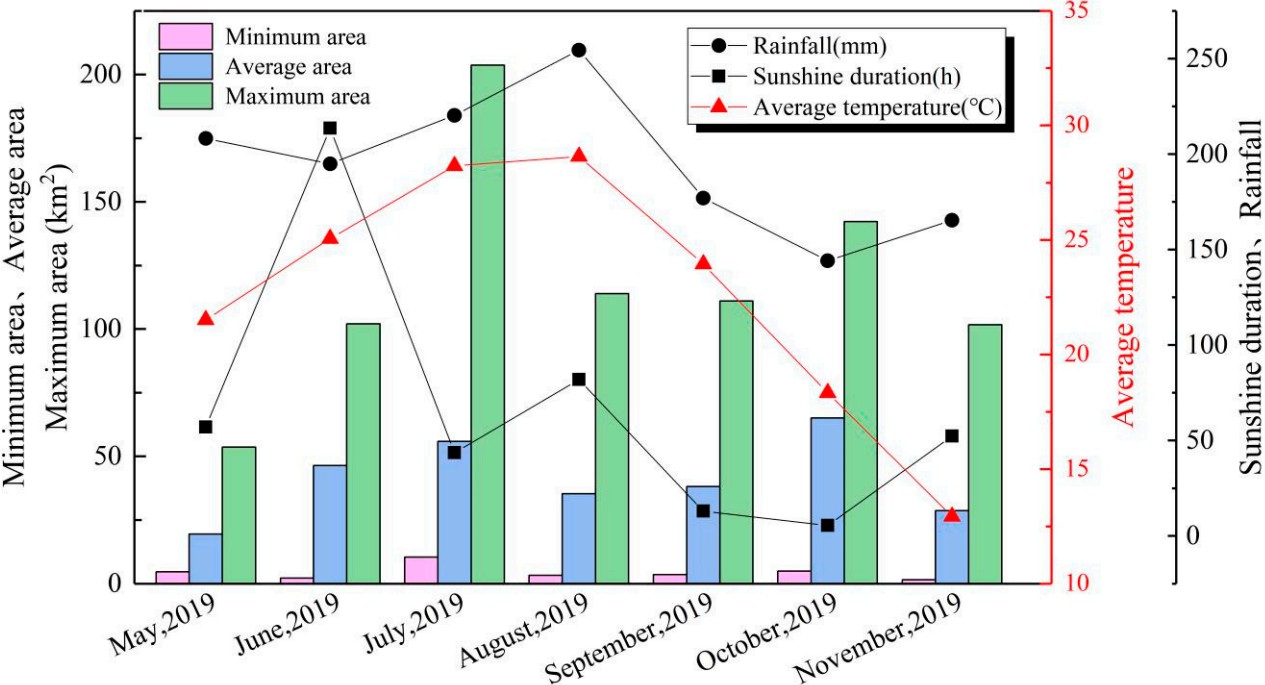

**Figure 13.** Chart of the minimum area, average monthly area, maximum area, average monthly temperature, sunshine hours, and precipitation of Chaohu Lake HABs.

The impact of wind speed on HAB showed a highly significant, positive correlation ($R^2 = 0.383$, $p < 0.01$). The wind direction map of Chaohu Lake in 2019 can be seen in Figure 14. A previous study revealed that when the average wind speed was larger than 3.8 m/s, the wind waves stirred the algal particles, causing them to sink, and reduced HAB concentration [72,73]. During the study period, only two days of HAB occurred with average wind speed greater than or equal to 3.8 m/s. The HAB area on August 12 was 4.8 km$^2$ (average wind speed of 4 m/s, average temperature of 28 °C), and the next day it was 113.94 km$^2$ (average wind speed of 1.5 m/s, average temperature of 29 °C). The solar radiation was similar, with sufficient sunshine hours (>9 h), but the HAB area was quite different. This indicated that the wind stirred up the algae particles so that the algae could not accumulate and sink, leading to a decrease in the HAB area. Moreover, appropriate wind speed and wind direction caused the HAB on the surface of Chaohu Lake to move toward the direction of the wind and accumulate. The results show that wind speed is an essential factor for the HAB outbreak and spread in Chaohu Lake. Prevailing winds in summer cause the shore water to converge on the northwest corner. The movement of water is not conducive to the material exchange on the surface of the flow field, which makes significant differences in the eutrophication pollution of algae of the whole lake [28]. Therefore, the frequency of HAB is the highest in the northwest of Chaohu Lake. There is

counter-clockwise circulation in the vicinity of Zhefu River in the eastern Chaohu Lake and clockwise circulation in the vicinity of Zhao River [28], which brings N, P, and other nutrients to the northeast of Chaohu Lake and near the middle of the lake, and the nutrients concentrate, resulting in many of HABs. Chaohu sluice, connecting the southeastern part of Chaohu Lake with Yuxi River, has a certain influence on the flow field near the eastern part of Chaohu Lake and plays a favorable role in the exchange of HAB with the outside.

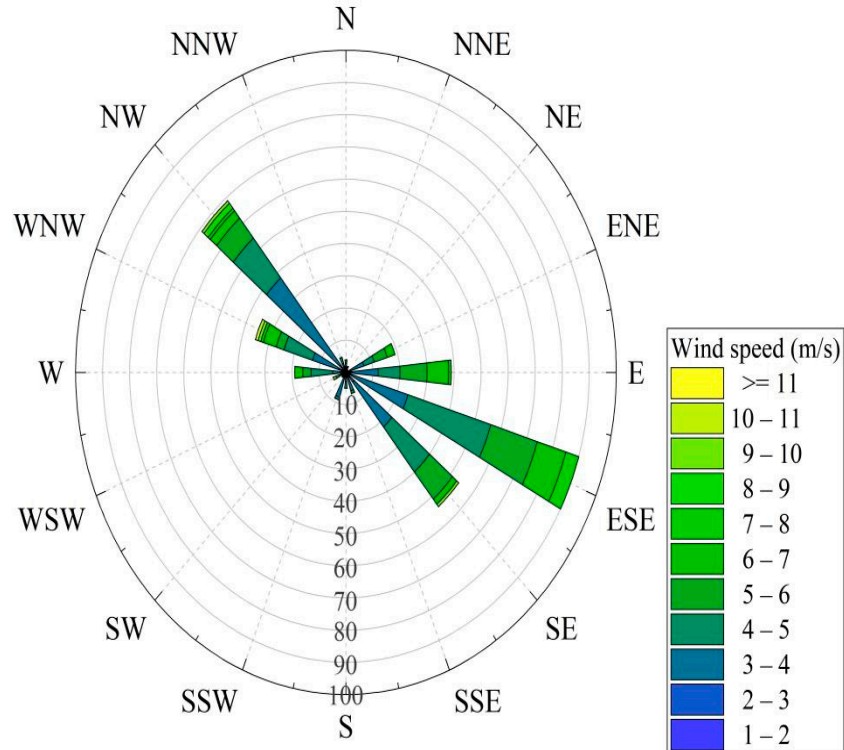

**Figure 14.** The wind direction map in Chaohu Lake 2019 (10, 20, 30, etc., indicate the number of days).

The average wind speed on 24 October 2019 was 1.5 m/s, which was less than the critical value (3.8 m/s) for algae aggregation and movement [74]. Additionally, the maximum wind speed was 3.8 m/s. As can be seen in the HAB distribution in Chaohu Lake detected by Terra and Aqua on October 24 (Figure 15a,c), the HAB in the central part of Chaohu Lake is gradually moving in the east–southeast direction, in line with the maximum wind speed direction 14 (that is, the west-northwest direction). On 8 November 2019, the maximum wind speed was 2.9 m/s, and the maximum wind speed direction was 3 (that is, a northeasterly). HAB areas in Chaohu Lake were 31.75 and 43.6 km$^2$, respectively, detected by Terra and Aqua. There was a low average wind speed (2 m/s) on Chaohu Lake on that day, which caused the algal particles to turn up and accumulate on the surface. The changes of HAB were also affected by wind waves, leading to the distribution location moving to the southwest (Figure 15b,d). Therefore, the multi-source remote sensing data can effectively monitor and reveal the diurnal change and development process of HAB.

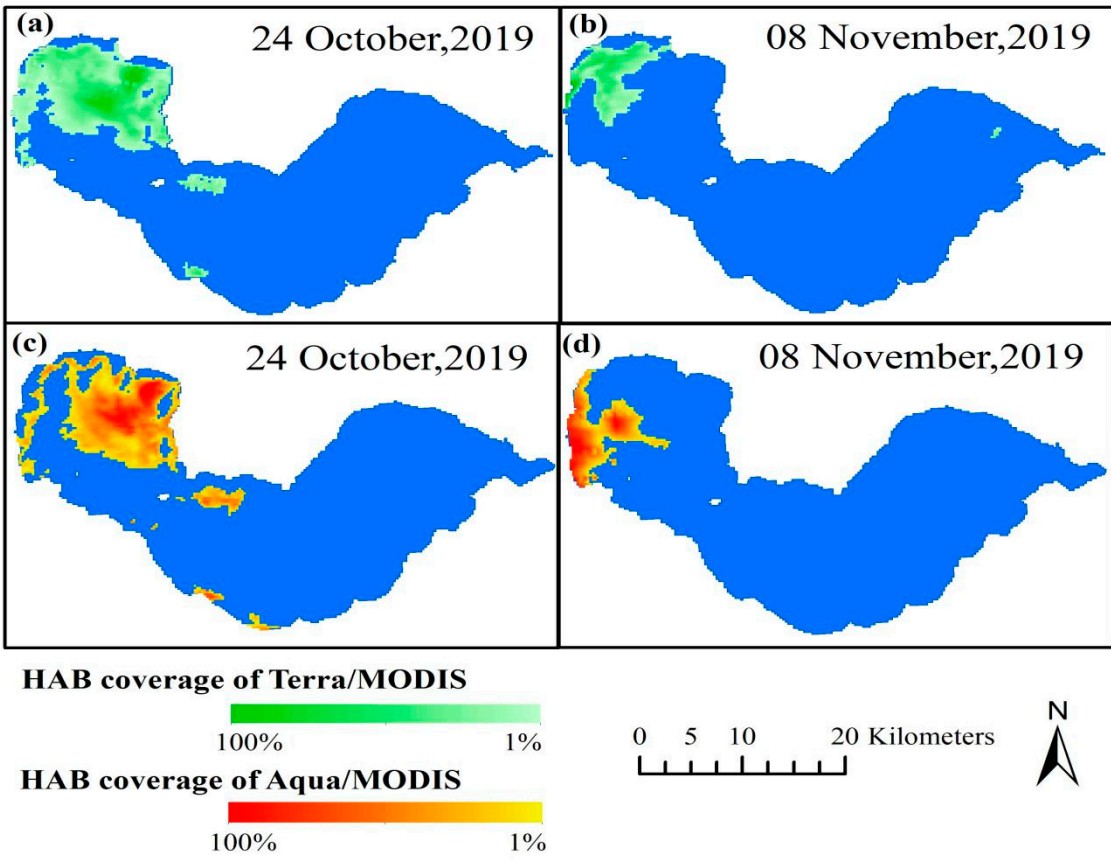

**Figure 15.** The intraday variations of HAB distribution: (**a**) Terra/MODIS image of the HAB distribution on 24 October, (**b**) Terra/MODIS image of the HAB distribution on 8 Novermber, (**c**) Aqua/MODIS image of the HAB distribution on 24 October, (**d**) Aqua/MODIS image of the HAB distribution on 8 Novermber.

### 5.2. Advantages of Multi-Source Satellite Remote Sensing

MODIS satellites with moderate spatial resolution have been widely used in monitoring HABs in large water bodies. However, the identification of HABs by moderate spatial resolution is limited in small inland water bodies or reservoirs and even has a large accuracy error. Due to the moderate spatial resolution, the boundary of a HAB identified by MODIS data is fuzzy, and the recognition ability of low-concentration HAB is low, leading to large uncertainties for monitoring HABs of a small inland lake. Sentinel-2 images, with a spatial resolution of 20 m, could significantly improve the identification accuracy and spatial details of HAB. For a concentrated outbreak area (Figure 16h), MODIS satellite has a relatively good performance in extracting HABs, but its ability to define the boundary of a HAB's area is weak. The error extraction rate is 40%, which is relatively high. Therefore, in the same timeframe, the extraction of HABs by combining multi-source data can verify and correct the extraction results of moderate-resolution images.

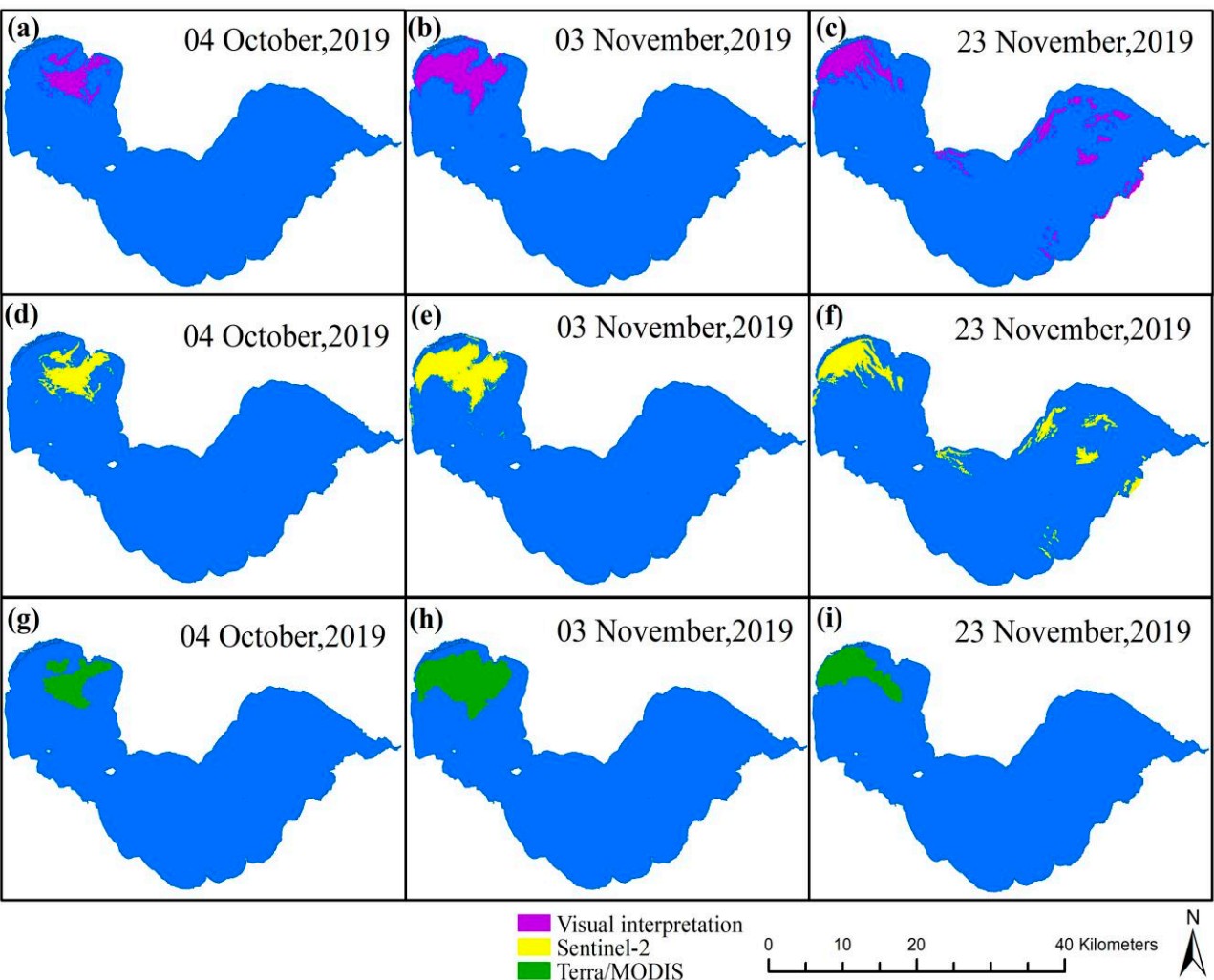

**Figure 16.** Comparison of the extraction results among Sentinel-2, Terra/MODIS, and visual interpretation: (**a**,**d**,**g**) Results of HAB extraction from Visual interpretation, Sentinel-2, Terra/MODIS on October 4, (**b**,**e**,**h**) Results of HAB extraction from Visual interpretation, Sentinel-2, Terra/MODIS on 3 November, (**c**,**f**,**i**) Results of HAB extraction from Visual interpretation, Sentinel-2, Terra/MODIS on 23 November.

In addition, remote sensing technology still makes it difficult to meet the requirements of high spatial-temporal resolution using a single satellite, especially for HABs with dramatic variations both spatially and temporally. To achieve both high spatial and high temporal resolution, multi-source satellite integration is an effective method to monitor the HABs in Chaohu Lake. Combined use of Terra/Aqua MODIS, Sentinel 2 MSI, and Landsat 8 OLI could provide more than three times per day monitoring of HAB, which is more efficient and accurate. For instance, parts of HAB information would be missed if only one satellite dataset was used; e.g., on 23 November 2019, some areas of HAB on the eastern part of Chaohu Lake would have been ignored by Terra image. By making full use of the advantages of multi-source images and monitoring the diurnal or long time scale changes of HAB in Chaohu Lake, they can learn from each other and make up for their shortcomings. Compared with single remote sensing data, more objective and accurate results were obtained.

## 6. Conclusions

Satellite remote sensing provides great potential to contribute significantly to the need for monitoring the HABs at a large scale; however, a multi-source remote sensing-based approach is preferred to achieve high temporal and spatial resolution observations of the

HABs, such as the integration of Terra/Aqua MODIS, Landsat 8 OLI, and Sentinel-2A/B MSI. With the advantage of the high temporal resolution, MODIS data are efficient in tracking the inter-monthly variations and distributions of HABs. In contrast, the integrated multi-satellite data provide the possibility to grasp the breakout and spread, especially the diurnal change of a given HAB, which is more objective and accurate than results from one single satellite's monitoring, as shown in the case of the Chaohu Lake. To obtain reliable HAB monitoring results, it is crucial to determine an appropriate HAB detection method considering the spectral characteristics of HABs and the band settings of different satellite sensors, and our study proved that NDVI is suitable for MODIS; NDVI and *FAI* combined for Landsat 8 OLI; and the NDVI and $\rho_{chl}$ combined for Sentinel-2 MSI data. Besides, analysis of driving forces of HAB, including environmental and meteorological factors of temperature, rainfall, sunshine hours, and wind, indicated that higher temperatures and light rain favored HAB. The wind is the main factor in boosting a HAB's growth. Multi-source remote sensing provides higher measurement frequency and more detailed spatial information on the HAB, particularly the HAB's long-short term variations. The results can be used as baseline data to evaluate the lake's HAB and water quality management in the future.

**Author Contributions:** Conceptualization, J.M. and S.J.; methodology, J.M., J.L., Y.H. and S.J.; software, J.M.; validation, J.L. and S.J.; formal analysis, J.M.; investigation, J.M.; resources, J.M. and S.J.; data curation, J.M.; writing—original draft preparation, J.M. and S.J.; writing—review and editing, J.L., W.S. and S.J.; visualization, J.M.; supervision, S.J.; project administration, S.J.; funding acquisition, S.J. All authors have read and agreed to the published version of the manuscript.

**Funding:** This work was funded by the Strategic Priority Research Program Project of the Chinese Academy of Sciences (grant number XDA23040100), the Startup Foundation for Introducing Talent of NUIST (grant number 2018R037), and the Jiangsu Province Distinguished Professor Project (grant number R2018T20).

**Institutional Review Board Statement:** Not applicable.

**Informed Consent Statement:** Informed consent was obtained from all subjects involved in the study.

**Data Availability Statement:** The data presented in this study are available on request from the corresponding website.

**Acknowledgments:** The authors would like to acknowledge the data provided by the National Aeronautics and Space Administration (NASA), European Space Agency (ESA), and Planet Labs.

**Conflicts of Interest:** The authors declare no conflict of interest.

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
