# Peer review of "Spatio-Temporal Variations and Driving Forces of Harmful Algal Blooms in Chaohu Lake: A Multi-Source Remote Sensing Approach"

_remotesensing, doi:10.3390/rs13030427_

Round 1
Reviewer 1 Report
L21 Please remove "Lndsat8" by "Landsat 8".
L102: Please provide GCS.
L126-L132: Please, give more details about number of scenes used for L8 and S2.
Figure 4: Replace "Sen2or" by "Sen2cor"
L174: How was grounds objects distributed? which decide did you use to measure then? Did you check all images?
L174-170. This paragraph is link only for S2 images?
Figure 5: Is this figure described in the text?
Ln 207 and 208. If you wrote NDVI acronym at line 205, please use it later.
Eq 1: Replace R by ρ
Ln 218 remove "light"
Ln 235: Replace B2, B3 and B5 by region names.
Ln 237-242: it is a result, please move it. In addition, you have to provide more information and explain it properly in method section.
L250-255: How were these control areas?, Did you check all of them?
Ln 262: You should write in past.
Ln 269. Your RMSE is equal to 4.52 km2. How did you get real values? if you measure on screen it is not correct.
Figure 7. Add line 1:1. You have extreme values that forces good results. Please provide more information about your adjustment like Cook's distance, normal distribution error test, etc.
Table 2. Sorry, I do not understand "obfuscation" matrix. In addition, why have you used cloudy image?
Ln 285-287. In my opinion, you are wrong using cloudy images.
Ln 333. Is it "Figure 9" correct?
Ln 338. Be clear, do you mean spatial resolution?
Author Response
Dear reviewer:
We would like to thank the editor for giving us the opportunity to modify our manuscript. We give our appreciations to the reviewers for their professional and thoughtful comments and suggestions on the manuscript. We have carefully taken their comments into consideration when preparing our revision, which has resulted in a new version of the manuscript that is clearer, more compelling, and broader.
Our replies to reviewers’ comments are attached below. We sincerely hope that the reply and the revisions can satisfy the editor and referees’ expectations.
Kind regards,
Jieying Ma

Reviewer 2 Report
The manuscript is well structured and includes a great deal of information. Sometimes the details make the reader get lost, so a review of the Materials and methods section is advised.
For example, table 1 and figure 2, it is not clear what they indicate, if the images used are those affected by cloudiness ... What does "revisit" mean? It is important that the starting data are clear because they are the basis of the article.
The Landsat bands are not expressed in nm but in micrometers.
It is not stated what DN and TOA mean. Table 2 is not well understood due to the accuracy data.
The authors must correct the misprints especially of the Greek letters.
Author Response

(The authors gave the same response as above.)

Reviewer 3 Report
This paper focused on the integration of the images of the several satellites for managing harmful algae distribution in the lake. I got a good impression with their effort for the combination of the satellite images in the methodologies.
Perhaps the first reviewer gave some comments to the introduction and Methods. I felt that authors replied to those comments sincerely and revised. So that, I will give some comments and revisions mainly in the chapters of Results and Discussion.
Page 14 Line 314 What is the definition of “Total HAB area” for each month? Icould not find the definition in the sentences.
Page 14 Line 315 “minimum” is not up to 50%
Page 14 Line 316<=>Line319
Ratio is up to 50% and area is only 53.69km2 in May,
Ratio is less than 25% and area exceed 100km2 from Jun to Nov
What are the meanings of these numbers?? The magnitude correlation is opposite. Perhaps that relates to the definition of total HAB area.
Page 15 Line 333 Figure 9 => Figure 11
Page 15 Line 336 repeating “on June 26”
Page 15 Line 338-341 I could not catch up the sentence logically. Why the difference of the shooting time and satellite tracing directions of Terra and Aqua related to the lower accuracy than Sentinel?
Page 15 Line 341-343 the sentence looks grammatically incorrect for me. In addition, if my guess to this sentence is correct, it sounds strange. The author wants to describe that the nutrients were transported by the surface current driven by the wind, then the bloom occurs. Perhaps the floating algae can be more easily transported by the wind driven current of wind itself. In fact, the author himself describe the following sentence in line 344. Totally, the reconsideration with the sentences Line 335-349 is required.
Page 17 Line 366&369 Average 2.4m/sec, Maximum >5m/sec Which velocity should be referred? In addition, that move is not transportation of the same aggregate of algae, just due to the sink of aggregate stirred up. This is the discussion for that in the center. On the other hand, how did the aggregate in the west (bigger than that in the center in Aqua image) behave? Figure 13 overlay the 2 images, sob that I could not distinguish the behavior of that aggregate from the image.
Page17 Line 373 Figure 12b=> Figure 11 g&i ?? or Figure 13
Figure 13 is not useful for understanding due to overlaying
Page17 Line 375 14”m/s”
Totally reconsideration of the sentences Line366-382. Focus is not clear.
The relationship with wind speed and direction should be moved to the chapter of discussion.
Page18 Figure 13 What the author want to express with this figure is not clear in the sentences. Need to explain how you draw these figures. I guess overlaying distribution in Terra images were indicated with green gradation and that in Aqua were with red-to-yellow. Such descriptions are needed in the sentence. Is the legend indication correct? Is this number percentage? The gradation color indicate what?
Page 18 Line 402 The effect of precipitation—
Not weak, not only dilution. Much rain in June transport the nutrients from the catchment area as the non-point source. The bloom in July with the highest maximum area is due to the inflow during June. Effect of Nutrients supply appears with the time lag, because controlling factor is temperature. Even with high conc. of nutrients, insufficient temp. regulates a blooming. That’s why I insist the temperature in Figure 14 later. I guess the large part of the precipitation in June is near to the end of the month. Higher temporal resolution of precipitation will give more detail discussion.
Page 19 Figure 14 better to prepare the axis of temperature for the range between 15-25or 30 degree. That include critical temp for blooming.
Page 19 Line 433 “Meteorological condition” includes wind condition. How about change the word from meteorological to solar radiation?
Page 19 Line 443&444 No indication of Zhefu river and Zhao river in any map.
Page 19 Line 427-448 discussion with wind is extended but wind data series are not shown in figures. the most important factor should be shown with magnitude and direction.
Page 20 Figure 15 Overlaying hide the distribution of sentinel and Terra.
Author Response

(The authors gave the same response as above.)

Reviewer 4 Report
Formal comments:
The English needs to be improved, there are many typing errors and the syntax is not very good. Some parts of the text are even confusing or incomprehensible due to poor English and could lead to misinterpretation.
Some parts of the text are highlighted in red without evident reason, why they are in red.
Figure 5 is quoted in the text (line 245) far behind the place, where the figure appears (line 199).
It is not necessary to provide formulas of a simple basic statistics in the text, for example RMSE and the other ones (formulas 5 or 6).
Figure 11 (line 351) is not quoted in the text. The quote "Figure 9" in the text (line 333) should be obviously "Figure 11".
There are many other errors, for example the Terra image is not shown in Figure 12d (line 356), but in Figure 12a (see line 363). The Landsat 8 image is not shown in Figure 12e (line 356), but in Figures 12b,d (see line 363)...
There are a number of citations in the article that are misleading. For example, the citation [40] does not refer to the Landsat 8 OLI sensor, as stated in the text (line 164) but to the Sentinel 2 MSI sensor. Moreover, the numeration of articles in the References chapter is incorrect (see lines 505, 561, 564, 566, 607, 649).
Factual comments:
The water environmental conditions in the study area could be described in more detail (lines 98 - 122). For example, typical nitrate nitrogen and phosphate phosphorus concentrations in lake water should be presented as well as other hydrological conditions (inflow, outflow). The average annual rainfall is reported to be 683 mm (line 112), but the maximum precipitation in June is reported to be 689 mm (line 142), which is confusing.
The Sentinel 2 MSI Panchromatic channel B8 with a very wide bandwidth range (Figure 4, line 152) is not usable for spectral analysis of the algae amount in water. The spatial resolution of used infrared bands of the Sentinel 2 MSI sensor is not 10 m per pixel (Table 1, line 133), but 20m.
The atmospheric corrections of the MODIS data (lines 164 - 168) should be described carefully in detail, the provided description is insufficient. The FLAASH atmospheric correction module was used for the Landsat OLI data and the SEN2COR radiometric and atmospheric correction module was used for the Sentinel 2 MSI data. These methods were developed especially for terrestrial data. In my opinion, the most modern DSF atmospheric correction method developed for the aquatic environment (Vanhellemont and Ruddick 2018) is much more suitable for the purposes of atmospheric correction of inland water satellite data and should be used in the work. The method is adapted to Landsat and Sentinel data using the ACOLITE module (Vanhellemont 2019), addresses some serious problems with dark spectrum fitting and significantly improves the accuracy of corrected data.
In the presented work, the HAB detection method (binary yes/no occurrence of HAB) relies on simple spectral calculations with thresholds that have been experimentally determined. The accuracy assessment was performed by visual inspection of false colour RGB composition of the satellite images. This is, in my view, an unacceptable method. The spectral data should be related to some measured physical or chemical quantity. Chlorophyll-a in green algae (and also in cyanobacteria) is the main pigment. Spectral indices used in the presented work detect exactly the spectral properties of chlorophyll. For example, a laboratory-determined concentration of chlorophyll-a in lake waters could be the quantity to which spectral data should be related. These methods are described massively in the literature, and even advanced mathematical methods (neural networks, machine learning methods, etc.) are also used to find the relationships of spectral and ground measured data. A binary assessment of the occurrence of HAB and a simple visual check make no contribution to contemporary knowledge.
In general, the impact of atmospheric conditions on the occurrence of algal blooms is well known and described in the literature, but the presented data (Figure 14, line 424) is too coarse to provide a better insight into the problem or at least a basic view.
References:
Vanhellemont Q., Ruddick K. (2018): Atmospheric Correction of Metre-Scale Optical Satellite Data for Inland and Coastal Water Applications. Remote Sensing of Environment, 216, pp. 586–597. doi:10.1016/j.rse.2018.07.015.
Vanhellemont Q. (2019): Adaptation of the Dark Spectrum Fitting Atmospheric Correction for Aquatic Applications of the Landsat and Sentinel-2 Archives. Remote Sensing of Environment, 225, pp. 175–192. doi:10.1016/j.rse.2019.03.010.
Author Response

(The authors gave the same response as above.)

Round 2
Reviewer 1 Report
Authors have taken into account my suggestions and it is adequate to be published.
Author Response
Dear reviewers:
Thank you for your affirmation of our manuscript. In further research, we will make up for the shortcomings and achieve perfection.
Kind regards,
Jieying Ma
Reviewer 4 Report
The authors have corrected a number of typing errors and other mistakes, but the text is not written very well and requires more syntactic editing to make it clearer.
The authors explained the choice of atmospheric correction method in a cover letter. I can understand the difficulties described and I think the explanation is acceptable, although the method provides corrected data with lower sensitivity for the purpose of spectral detection in surface waters. But I think some of the essential points of this explanation should be mentioned in the Methodology chapter.
I agree that the detection of HAB in a simple yes/no mode makes sense for quick evaluation. On the other hand, these techniques have been well known for a long time, and satellite imagery is used together with aerial data in the standard way commonly used for these purposes. Since the work makes no further contribution to contemporary knowledge and is based on techniques well known and used over a long period, I am in the unfortunate position of having to reject the paper.
Author Response
Dear reviewers:
Our replies to reviewers’ comments are attached below. We sincerely hope that the reply and the revisions can satisfy the editor and referees’ expectations.
Kind regards,
Jieying Ma

This manuscript is a resubmission of an earlier submission. The following is a list of the peer review reports and author responses from that submission.
Round 1
Reviewer 1 Report
Manuscript: Spatio-temporal variations and driving forces of harmful Algal Blooms in Chaos Lake: a multi-source remote sensing approach
This manuscript is aligned with the topics of Remote Sensing, however it needs changes to be accepted.
- Line 14: What do you mean whit "one sensor"?, do you mean EOS?
- Line 15: "urgently", it sounds bold.
- Line 19: Again you wrote "sensors", I understand what you mean but it is confused.
- Introduction: In the abstract you wrote about environmental and meteorological factors, however you did not write anything at all about them, moreover, there are no references in this section about these factors. Please write about it.
- Line 116: Please include coordinate reference system.
- Line 133: Which MODIS' product did you use?
- Line 135: Which software did you use for MODIS scene processing?
- Line 145 and 146: Please, replace "is" by "was"
- Figure 2: X-axis is quite difficult to read. In addition, I recommend to replace singular graph by 4 graphs.
- Line 170: What is measured results?
- Section 3.1. How did you remove clouds in scenes?
- Section 3.1. did you analyze correlation about different EOS products?
- Line 177: "first and second bands" is confused.
- Line 177: Did you use 250, 500 or 1000 spatial resolution bands from MODIS?
- Line 188: Replace "is" by "was".
- Line 217: How many scence did you validate? Please, explain how did you validate.
- Figure 3: It is evident there are clouds in your scene, how did these clouds affect to your results?
- Figure 5: Which sub-figure is from MODIS and Sentinel? Caption figure indicates that Sentinel has more data than MODIS, is it correct?
- Section 3.3. How did you check and validate your visual interpretation?
- Figure 7: I do not understand properly this figure.
- Line 313. Replace "Tarra" by "Terra".
- Figure 8: Can you explain why Terra and Aqua Modis results are different?
- Line 318: What do you mean with "quality"?
- Figure 9. Please, include Terra/MODIS results on 03 august, 2019.
- Section 5.1. It is a discussion section and you are showing correlation between factors. You have to provide more information.
- You should write conclusion section. it is like a little summary of your research.
Author Response

(The authors gave the same response as above.)

Reviewer 2 Report
I think the subject is quite interesting and that the research carried out is quite impressive considering the amount of data processed. However, the manuscript needs serious improvements: 1) the authors need to provide clear and solid theoretical background regarding their choices of methods; 2) the paper structure does not help. There is a mix of methodological content amongst the results. Sometimes, some results are presented without a clear connection to a methodological step. Therefore, the manuscript needs a profound revision before acceptance. However, I encourage the authors to proceed because the topic is very relevant and of general interest

Author Response

(The authors gave the same response as above.)

Round 2
Reviewer 1 Report
In my opinion manuscript is adequate to be published.
Reviewer 2 Report
A think the main aspect is that the method as far as I could understand is not well described. Important issues such as the approach used for assessing mapping accuracy, for instance, were not well described. It is also not clear how the author computed the monthly frequency of the bloom distribution based on a variable number of images.bacuese the number of samples were variable and small ( 6 to 18) provided that Acque and Terra images of the same day are considered independent samples. Another important issue which was not addressed at all is how glint could disrupt the different index used to map the blooms. More over, it is not clear how does the authors circunvent spatial changes on the cloud cover, the presence of cloud cover at different heigths which t even not covernig the water body, could increase the sky reflectance and disrupt the threshold efficient from one scene to the other.
.
